# Discovering Knowledge Deficiencies of Language Models on Massive Knowledge Base

**Linxin Song**[α]**, Xuwei Ding**[β]**, Jieyu Zhang**[γ]**, Taiwei Shi**[α]**, Ryotaro Shimizu**[θ]**,**
**Rahul Gupta**[ε]**, Yang Liu**[ε]**, Jian Kang**[ζ]**, Jieyu Zhao**[α]
[α]University of Southern California, [β]University of Wisconsin–Madison,
[γ]University of Washington, [θ]ZOZO Research, [ε]Amazon, AGI, [ζ]University of Rochester

 **Code:**  github.com/uscnlp-lime/SEA

## Abstract

Large language models (LLMs) possess impressive linguistic capabilities but often fail to faithfully retain factual knowledge, leading to hallucinations and unreliable outputs. Understanding LLMs' knowledge deficiencies by exhaustively evaluating against full-scale knowledge bases is computationally prohibitive, especially for closed-weight models. We propose stochastic error ascent (SEA), a scalable and efficient framework for discovering knowledge deficiencies (errors) in closed-weight LLMs under a strict query budget. Rather than naively probing all knowledge candidates, SEA formulates error discovery as a stochastic optimization process: it iteratively retrieves new high-error candidates by leveraging the semantic similarity to previously observed failures. To further enhance search efficiency and coverage, SEA employs hierarchical retrieval across document and paragraph levels, and constructs a relation directed acyclic graph to model error propagation and identify systematic failure modes. Empirically, SEA uncovers 40.7× more knowledge errors than Automated Capability Discovery and 26.7% more than AutoBencher, while reducing the cost-per-error by 599× and 9×, respectively. Human evaluation confirms the high quality of generated questions, while ablation and convergence analyses validate the contribution of each component in SEA. Further analysis on the discovered errors reveals correlated failure patterns across LLM families and recurring deficits, highlighting the need for better data coverage and targeted fine-tuning in future LLM development.

## 1 Introduction

Large language models (LLMs) are pretrained on vast corpora, including comprehensive knowledge sources such as Wikipedia. Despite this extensive exposure, LLMs often fail to retain or accurately reproduce factual information, resulting in misinformation and hallucinations (Gekhman et al., 2023; Zhang et al., 2023; Manakul et al., 2023; Jiang et al., 2024; Yu et al., 2024). For instance, LLM mistakes France's capital as Berlin (Simhi et al., 2024), or may fabricate a plausible-looking citation by attributing a fictitious paper to a real researcher in the relevant domain (Merken, 2025). These knowledge deficiencies pose significant challenges for downstream applications, particularly in high-stakes domains like healthcare, law, and scientific research, where factual accuracy is paramount.

To pinpoint knowledge deficiencies in LLMs, researchers typically construct static knowledge-intensive benchmarks for various domains (Chen et al., 2021; Hendrycks et al., 2021b;a; Song et al., 2023; Su et al., 2024; Yue et al., 2024; Li et al., 2024b; Yang et al., 2024; Phan et al., 2025), and use them to reflect the level of LLM's knowledge. However, considering the massive amount of knowledge that human society has accumulated, it is infeasible to curate static benchmarks that cover all of it and test LLMs against it. Therefore, the need arises for a versatile approach that can automatically uncover LLM's knowledge deficiencies

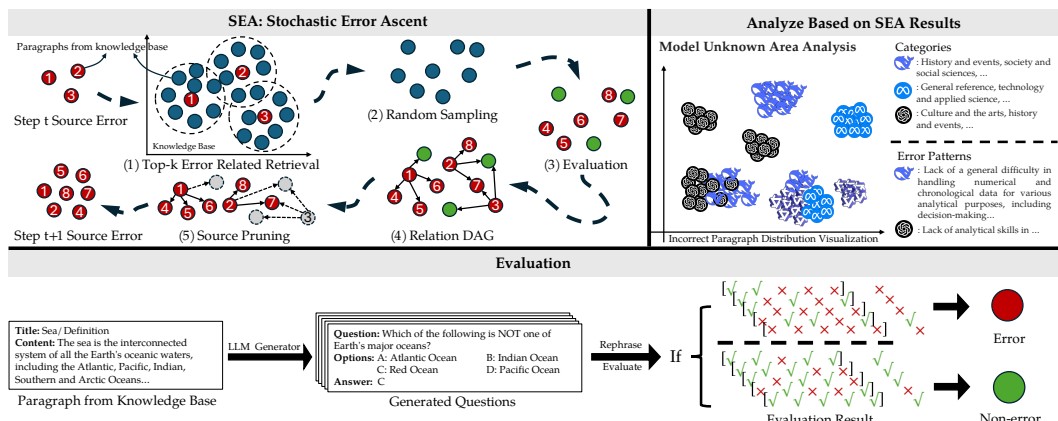

**Figure 1:** Overall workflow of stochastic error ascent (SEA). We search for a closed-weight model's unknown knowledge iteratively from a given knowledge base until we reach the budget. The result from SEA can be further used to analyze the model's unknown categories and error patterns.

from a massive knowledge base, ideally discovering as many deficiencies as possible within a limited evaluation budget for efficiency and scalability. Moreover, it should be applicable to closed-weight LLMs, given their widespread adoption for downstream tasks.

To this end, we propose stochastic error ascent (SEA, Fig. 1), a scalable framework for uncovering knowledge deficiencies in LLMs under a query budget. Exhaustively querying a massive knowledge base is infeasible, so we iteratively select subsets that are likely to induce new model errors by formulating the task as a stochastic optimization problem. At each step, SEA approximates the per-step optimal subset by retrieving samples most similar to prior errors, leveraging the observation that model failures often exhibit shared characteristics (Li et al., 2024a). To enhance efficiency, we adopt a hierarchical strategy—finding the samples first at the document level and then at the paragraph level. We construct a relation directed acyclic graph (relation DAG) that traces source-target error dependencies and prunes low-impact nodes based on cumulative errors, helping discover systematic weaknesses.

We conduct extensive quantitative and qualitative evaluations of SEA over eight commonly used LLMs. Compared to two baselines—Automated Capability Discovery (ACD, Lu et al. (2025)) and AutoBencher (Li et al., 2024a)—SEA uncovers 40.7× more errors than ACD and 26.7% more than AutoBencher, while reducing cost per error by 599× and 9×, respectively. Human evaluation over 1,000 randomly sampled questions confirms a 100% pass rate, i.e., all the model-generated questions in SEA are reliable. SEA exhibits consistent error discovery across steps, with all components contributing comparably, as shown in convergence and ablation studies. Correlation analysis reveals strong intra-family model alignment, except for `o1-mini`. Most models perform well on `gpt-4o` and `gpt-4o-mini` optimal subsets but struggle on `DeepSeek-V3` ones. Error visualizations reveal two overlapping clusters, highlighting model-specific weaknesses and informing future data collection strategies.

## 2 Related work

**Dynamic benchmarking.** Static benchmarks (Chen et al., 2021; Hendrycks et al., 2021b;a; Song et al., 2023; Liang et al., 2024; Yue et al., 2024; Li et al., 2024b; Yang et al., 2024; Wang et al., 2024b; Jain et al., 2024; Shi et al., 2024; 2025; Phan et al., 2025) suffer from data leakage and high building costs. Dynamic frameworks address this via automated, evolving data generation. Dynabench (Kiela et al., 2021) integrates human-model interaction, while Task Me Anything (Zhang et al., 2024a) enables scalable, user-driven evaluation. Recent work—DyVal (Zhu et al., 2024a;b), UniGen (Wu et al., 2024), and Benchmark Self-Evolving (Wang et al., 2024a)—employs probing agents and multi-agent systems for iterative refinement. AutoBencher (Li et al., 2024a) casts benchmark construction as optimization to surface model flaws but remains limited by static templates or annotated data (Huang et al.,

2025). In contrast, SEA adopts a fully adaptive, error-driven probing strategy using relation DAG to uncover failures.

**Model behavior understanding.** We investigate factual knowledge modeling in LLMs, focusing on distinguishing correct from plausible-but-false outputs—a challenge rooted in their latent factual encoding (Petroni et al., 2019). Techniques such as gradient-guided prompt generation (Shin et al., 2020), systematic prompt design (Jiang et al., 2020), and tools like Automated Capability Discovery (Lu et al., 2025) further expand the evaluation scope. EvalTree (Zeng et al., 2025) further reveals vulnerabilities via hierarchical capability trees. However, reliance on hand-crafted prompts or constrained queries limits their efficacy in surfacing nuanced misinformation.

## 3 Deficiency Discovery for Large Language Models

Consider we have a massive knowledge base $\mathcal{K}$, a closed-weight foundation model $f_{\text{close}}$, and a given budget $C$. $\mathcal{K}$ includes $N$ documents, each with an abstract and $M_i$ paragraphs $p$, i.e., $\mathcal{K} = \{p_j^{(i)} | i = 1, ..., N; j = 1, ..., M_i\}$. **Our goal** is to find an optimal paragraph subset $\hat{\mathcal{S}} = \left\{p_1, \cdots, p_{|\hat{\mathcal{S}}|}\right\}$ from $\mathcal{K}$ that can maximize the error $T_{\hat{\mathcal{S}}}(f_{\text{close}})$ of the closed-weight language model $f_{\text{close}}$ under the budget $C$ for $f_{\text{close}}$ (e.g., the price for all tokens or a number of API calls). Specifically, we aim to solve the following optimization problem:

$$\underset{\mathcal{S} \subset \mathcal{K}}{\arg\max} \, T_{\mathcal{S}}(f_{\text{close}}) \quad \text{s.t.} \sum_{|g(\mathcal{S})|} \text{cost}(f_{\text{close}}) < C, \tag{1}$$

where

$$T_{\mathcal{S}}(f_{\text{close}}) = \frac{1}{|g(\mathcal{S})|} \sum_{p_j^{(i)} \in \mathcal{S}} \left( \sum_{(x,y) \in g\left(p_j^{(i)}\right)} \mathbb{1}_{\left[f_{\text{close}}(x) \neq y\right]} \right). \tag{2}$$

Eq. (2) denotes the average error of the closed-weight model $f_{\text{close}}$, where $(x, y)$ is the multiple-choice question-answer pairs generated from each $p_j^{(i)}$ by prompting a question generator LLM $g(\cdot)$, and $\mathbb{1}_{[\cdot]}$ is an indicator function. To avoid the accidental error triggered by the prompt sensitivity of $f_{\text{close}}$, we rephrase each question multiple times and generate semantically equivalent variants. To simplify, we use $T_{\mathcal{S}}$ for $T_{\mathcal{S}}(f_{\text{close}})$ in the later sections.

### 3.1 Stochastic Error Ascent

To achieve our optimization objective in Eq. (1), we propose a multi-step stochastic error ascent (SEA) algorithm. Generally, SEA iteratively performs error-based subset updates with hierarchical error-related retrieval until we reach the budget $C$. During the error-related retrieval, we construct a relation directed acyclic graph (relation DAG) with the source errors as nodes and error relations across each timestamp as edges. We further prune the source error by tracking the quality of the per-step source error. We present the overall algorithm in Alg. 1.

**Error-based subset update.** Given a large $\mathcal{K}$, it is nearly impossible to let $f_{\text{close}}$ go through all questions generated from $p \in \mathcal{K}$ to find an optimal set of paragraphs $\hat{\mathcal{S}}$ that can achieve the Eq.(1) under a limited budget. Therefore, we consider updating $\mathcal{S}$ iteratively. At timestamp $t$, we find a batch of candidate paragraphs $\mathcal{P}_t = \left\{p_t^{(1)}, \cdots, p_t^{(|\mathcal{P}_t|)}\right\}$ from $\mathcal{K}$ that can maximize the probability of $T_{\mathcal{S}_t \cup \mathcal{P}_t} > T_{\mathcal{S}_t}$, which can be represented as:

$$\mathcal{S}_{t+1} = \mathcal{S}_t \cup \underset{\mathcal{P}_t \subset \mathcal{K}}{\arg\max} \, \Pr\left(T_{\mathcal{S}_t \cup \mathcal{P}_t} > T_{\mathcal{S}_t}\right). \tag{3}$$

We solve Eq.(3) by focusing on paragraphs that resemble the error-inducing examples. Specifically, we target regions in $\mathcal{K}$ that are likely to further challenge the model, resulting

---

**Algorithm 1:** Stochastic Error Ascent

---

**Input:** Knowledge base $\mathcal{K}$, closed-weight model $f_{\text{close}}$, budget $C$, random initial paragraph set $B$, threshold $\xi$ and $\gamma$.

**Output:** Optimal paragraph set $\hat{\mathcal{S}}$

1 **Initialization:** $t \leftarrow 1$, $\text{cost} \leftarrow 0$, $\mathcal{S}_t \leftarrow \varnothing$, $\mathcal{P}_{\text{source}}^{(t)} \leftarrow \varnothing$

2 **while** $\text{cost} < C$ **do**

3     **if** $t = 1$ **then**

4        $E \leftarrow B$                       $\triangleright$ No source error at step 1; use random initial batch.

5     **else**

6        $E \leftarrow \text{FindSim}(\mathcal{K}, \mathcal{P}_{\text{source}}^{(t)})$       $\triangleright$ Eq.(4): Sample an error related candidate batch.

7     $\mathcal{S}_{t+1} = \mathcal{S}_t \cup E$                              $\triangleright$ Update the target subset.

8     $\mathcal{P}_{\text{source}}^{(t+1)} \leftarrow \mathcal{P}_{\text{source}}^{(t)} \cup \left\{ p \in E \mid T_{\{p\}} > \xi \right\}$         $\triangleright$ Update source error set.

9     $\mathcal{K} \leftarrow \mathcal{K} \setminus \left\{ p \in E \mid T_{\{p\}} > \xi \right\}$     $\triangleright$ Remove new source error from $\mathcal{K}$ to avoid loop.

10     $\mathcal{G}_{t+1} \leftarrow \left( \mathcal{P}_{\text{source}}^{(t+1)}, \mathcal{E} \left( \mathcal{P}_{\text{source}}^{(t)}, \mathcal{P}_{\text{source}}^{(t+1)} \right) \right)$           $\triangleright$ Update relation DAG.

11     $\mathcal{P}_{\text{source}}^{(t+1)} \leftarrow \mathcal{P}_{\text{source}}^{(t+1)} \setminus \left\{ p \in \mathcal{P}_{\text{source}}^{(t+1)} \mid \pi_{\mathcal{G}_{t+1}}(p) < \gamma \right\}$       $\triangleright$ Eq.(5): Pruning $\mathcal{P}_{\text{source}}$.

12     $\text{cost} \leftarrow \text{cost}(T_E(f_{\text{close}}))$          $\triangleright$ Update cost. We did one-time inference on $E$.

13     $t \leftarrow t + 1$

14 **return** $\hat{\mathcal{S}}$

---

in retrieving an error-related batch $E$ that is semantically similar to a set of source error paragraphs: $\mathcal{P}_{\text{source}}^{(t)} \subset \bigcup_{i=1}^{t} \mathcal{S}_i$. Here, $\mathcal{P}_{\text{source}}^{(t)}$ comprises paragraphs for which the LLM exhibits a high error rate (i.e., $T_{\{p\}} > \xi$ for all $p \in \mathcal{P}_{\text{source}}^{(t)}$, with $\xi$ being a predefined error threshold). Practically, we retrieve the error-related batch $E$ by ranking candidates from $\mathcal{K}$ based on their semantic similarity to $\mathcal{P}_{\text{source}}^{(t)}$, using tools such as Sentence Transformers (Reimers & Gurevych, 2019), and selecting the top-$k$ candidates. Given a Sentence Transformer $f_s : p \to \mathbb{R}^d$, where $d$ is the embedding dimension, we can define the error-related batch as:

$$E = \text{FindSim}(\mathcal{K}, \mathcal{P}_{\text{source}}^{(t)}) = \left\{ p_c \mid p_c \in \bigcup_{p_s \in \mathcal{P}_{\text{source}}^{(t)}} \text{Top}_k \left( \frac{f_s(p_s) \cdot f_s(\mathcal{K})}{\|f_s(p_s)\| \|f_s(\mathcal{K})\|} \right) \right\}, \quad (4)$$

where $f_s(\mathcal{K}) = [f_s(p_1), \cdots, f_s(p_{|\mathcal{K}|})]^\top$. To improve the efficiency of calculating Eq.(4), we perform hierarchical retrieval from document to paragraph levels. We first pre-process the embeddings for the abstract of all documents $d_i \in \mathcal{K}$ as $\mathcal{D}_{\text{abs}}$, where $d_i$ contains a set of paragraphs that can be represented as $\{p_{1:M_i}^{(i)}\}$. We then retrieve a set of candidate error-related documents $\mathcal{D}_c$ by performing Eq.(4) between $\mathcal{D}_{\text{abs}}$ and $\mathcal{P}_{\text{source}}$, i.e., $\mathcal{D}_c = \text{FindSim}(\mathcal{D}_{\text{abs}}, \mathcal{P}_{\text{source}})$. Finally, we retrieve error-related batch $E$ by comparing the paragraphs in document $\mathcal{D}_c$ and source errors, i.e., $E = \text{FindSim}(D_c, \mathcal{P}_{\text{source}})$. In this way, we finish the hierarchical retrieval.

**Relation DAG construction and source pruning.** To identify systematic issues in $f_{\text{close}}$, such as flaws localized within specific regions of the knowledge base, we construct a relation DAG $\mathcal{G} = (\mathcal{P}_{\text{source}}^{(t)}, \mathcal{E}(\mathcal{P}_{\text{source}}^{(t)}, \mathcal{P}_{\text{source}}^{(t+1)}))$. $\mathcal{G}$ is constructed by linking each paragraph $p \in \mathcal{P}_{\text{source}}^{(t)}$ to its top semantically similar error-inducing paragraphs in $\mathcal{P}_{\text{source}}^{(t+1)}$, based on the hierarchical error-related retrieval described above, forming directed edges that represent potential error propagation paths. We then assess the quality of $\mathcal{P}_{\text{source}}$ based on *cumulative error*. We define the *cumulative error* $\pi_{\mathcal{G}}(p)$ for paragraph $p \in \mathcal{P}_{\text{source}}$ as the average error across its descendants' error:

$$\pi_{\mathcal{G}}(p) = \frac{1}{|\text{Desc}_{\mathcal{E}}(p)|} \sum_{v \in \text{Desc}_{\mathcal{E}}(p)} T_{\{v\}}, \quad (5)$$

where $\mathrm{Desc}_{\mathcal{E}}(p)$ denotes all descendants of $p$ that can be reached via the edge space $\mathcal{E}$. We **prevent loops** in $\mathcal{G}$ by removing new source errors in each step from $\mathcal{K}$ (line 11 in Alg. 1). We then perform a threshold filter according to $\pi_{\mathcal{G}}(p)$ at each step to prune $\mathcal{P}_{\text{source}}$.

## 4 Comparing Stochastic Error Ascent with Baselines

**Knowledge base details.** We collect a large-scale English-based knowledge base from Wikipedia (Wikipedia contributors, 2004), comprising 7.1M documents and 28.8M paragraphs across 13 top-level categories with hierarchical subcategories. Each Wikipedia page is a document with a preprocessed abstract, and we map sections in the document as paragraphs. To enable efficient retrieval, we pre-embed each page's title and abstract using a Sentence Transformer and further process the embeddings via FAISS (Douze et al., 2024) with vector quantization and an inverted file system. Additionally, we convert HTML content to markdown to support on-the-fly paragraph-level retrieval.

**Evaluation protocol.** We use an LLM generator (gpt-4o) to generate and rephrase questions for each paragraph. Specifically, the LLM generator will take a paragraph as input and output a list of questions in JSON format. We perform the human evaluation of the generated questions and put the quality analysis in Sec. 5. We compute accuracy for the questions answered by all models by comparing the ground truth answer provided by the LLM generator and the answer from the testee models. We formulate all the questions with the same template for all testee models.

**Implementation details.** For $f_{\text{close}}$ selection, besides widely used closed-weight models, we also simulate open-weight models in the closed-weight setting to expand the scope of evaluation. Specifically, we evaluate GPT-series models: gpt-4o (OpenAI et al., 2024), gpt-4o-mini (OpenAI, 2024a), and o1-mini (OpenAI, 2024b), and DeepSeek-series models: Deepseek-V3 (Liu et al., 2024), Deepseek-R1, and R1-Distill-Llama-70B (DeepSeek-AI et al., 2025). We also include two widely used models hosted on DeepInfra[1]: Qwen2.5-72B-Instruct (Qwen et al., 2025) and Llama-3.3-70B-Instruct (Grattafiori et al., 2024). We adopt mGTE (Zhang et al., 2024c) as our Sentence Transformer model for hierarchical retrieval. We set the decoding parameters to temperature 0.1 and top-$p$ 0.9 for deterministic responses. Thresholds for source error judgment and source error pruning are set to $\xi = \gamma = 0.5$. We use the average accuracy across the 25 generated questions for source error judgment. We set the $k$ as 50 in Eq.(4) and randomly select 40 paragraphs from the retrieved paragraphs as the error-related batch $E$. For the initial batch $B$, we uniformly retrieve 40 paragraphs from 13 predefined Wikipedia categories.

**Baselines and budget settings.** Research on automatically generating questions to self-discover misinformation and vulnerabilities in LLMs remains limited. We select two closely related methods, Automated Capability Discovery (**ACD**, Lu et al. (2025)) and **Auto-Bencher** (Li et al., 2024a), as baselines to compare the ability of error discovery and cost. ACD is an automated method that leverages LLM's internal knowledge to self-discover some surprising successes and failures, while AutoBencher takes an input topic and retrieves related pages iteratively from a given knowledge base to build a challenging benchmark. Both methods are motivated by the need to overcome the high demand for human labor and the' limited ability of traditional evaluations, aiming to automatically explore models' capabilities and shortcomings. In our experiments, we use gpt-4o as the generator model for task/benchmark generation for ACD and AutoBencher. We let ACD take one of the Wikipedia topics as seed tasks for task generation, and let AutoBencher use all categories as interest topics and generate 13 benchmarks, including 2,000 questions in total. We adopt two different budget settings for SEA according to the features of ACD and AutoBencher. Specifically, we set the budget as 20,000 API calls for $f_{\text{close}}$ plus the cost for QA generation when comparing with ACD, and the budget as AutoBencher to generate 13 benchmarks when comparing with AutoBencher.

---

[1] https://deepinfra.com/

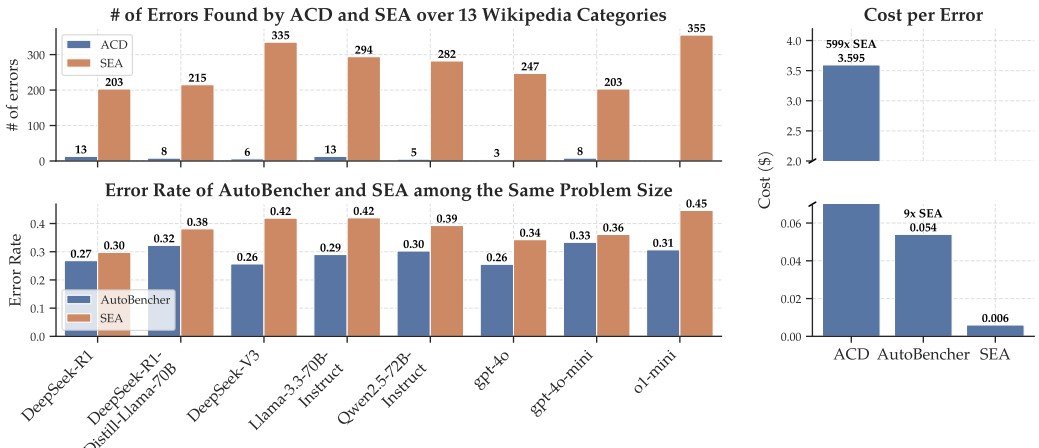

**Figure 2:** Comparison of errors discovered by ACD, AutoBencher, and SEA. We compare ACD with SEA among the same budget while comparing AutoBencher among the same question size. For ACD, we summarized the number of failed tasks, and for SEA, we summarized the number of source errors. We let AutoBencher create 13 benchmarks, each of which takes one of the Wikipedia categories as an interesting topic. We let SEA search the same number of questions according to each model. `o1-mini` failed on ACD due to the violation of the prompt usage policy from OpenAI.

## 4.1 Comparing with ACD

We first compare the number of errors ACD and SEA discovered among the same budget, including question generation and testee inference. Specifically, we let ACD start searching from a free-style handwritten task conditioning on one of the 13 general Wikipedia categories and summarize the error tasks discovered by ACD. Error tasks and source errors both reflect LLM misinformation in a category. Therefore, we summarize the comparison results between error tasks and source errors found by ACD and SEA, respectively, for each model across 13 categories in Fig. 2. We can observe a significant gap between the number of errors found by SEA and ACD. SEA can discover at most 55.83 times as many as the errors of ACD on `DeepSeek-V3` model. The main reason is that ACD does not rely on an external knowledge base but relies only on LLMs' internal knowledge, which requires more budget to guide LLMs into deeper regions of the error space.

## 4.2 Comparing with AutoBencher

We compare the error rates of questions generated by SEA and AutoBencher. Specifically, we let AutoBencher generate 13 benchmarks, each corresponding to a distinct Wikipedia category as the input topic. We then concatenate all benchmarks as one (resulting in 2,000 questions) and evaluate all models based on it. We compare the error rate between the AutoBencher benchmark and the error rate on an equal number of questions generated by SEA. The results are summarized in Fig. 2. As shown in Fig. 2, SEA outperforms AutoBencher across all evaluated models in terms of error rate. Specifically, on `DeepSeek-V3`, SEA achieves an error rate of 0.42, substantially higher than AutoBencher's 0.26, indicating a 61.5% relative improvement in error detection capability. Similarly, for `Llama-2-70B-Instruct` and `Qwen2-72B-Instruct`, SEA records error rates of 0.42 and 0.39, surpassing AutoBencher by margins of 13% and 30%, respectively. The average error rate across all models is 0.38 for SEA versus 0.30 for AutoBencher, reflecting a 26.7% increase in error detection efficiency.

## 5 Analyzing Stochastic Error Ascent

**Validating with human evaluation.** In this study, we utilize `gpt-4o` as the question-answer generator. We assess the quality of its outputs through a human evaluation involving

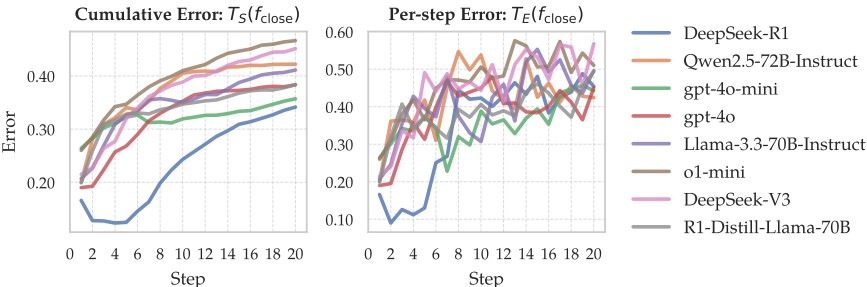

**Figure 3:** Per-step error $T_E(f_{\text{close}})$ and cumulative error $T_{\mathcal{S}}(f_{\text{close}})$ for each model. We observe that the errors of all models are positively related to step, indicating SEA can gradually and continually find the model's knowledge deficiencies from the knowledge base.

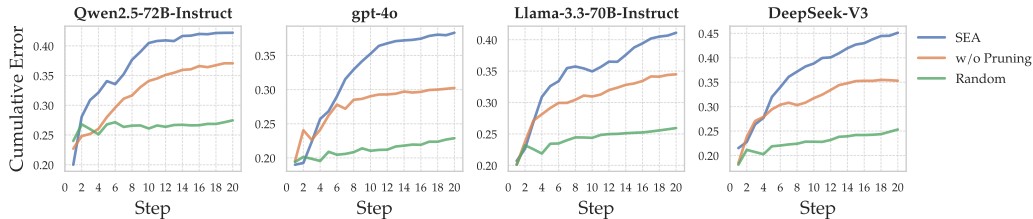

**Figure 4:** Ablation studies on the component contribution of SEA. We compare SEA with its two variants: without source pruning (i.e., pass the lines 10 and 11 in Alg. 1) and random selection (i.e., pass the lines 9, 10, and 11 in Alg. 1). We observe that each component contributes equally to SEA.

five college-level students, who verify the truthfulness of the generated answers by cross-referencing them with corresponding paragraphs in the knowledge base, mirroring the input provided to the generator. From 1,000 randomly selected questions generated across 20 steps, our evaluation achieved a 100% human pass rate, confirming that all answers were both present and correct within the associated paragraphs, consistent with the results in Li et al. (2024a), which also uses LLM as a question generator with Wikipedia documents.

**Convergence analysis.** To analyze the convergence behavior of SEA, we conduct an empirical study, given the inaccessibility of internal activations in closed-weight models during the search process. We track both the cumulative error $T_{\mathcal{S}}(f_{\text{close}})$ and per-step error $T_E(f_{\text{close}})$ over 20 iterations of SEA (Fig. 3). For cumulative error (left), we observe a steep initial increase followed by a plateau across all models, indicating that SEA rapidly identifies high-impact errors early in the search, then gradually uncovers subtler or less frequent failure modes. Notably, models such as `o1-mini` and `R1-Distill-Llama-70B` exhibit higher peak cumulative errors, while `DeepSeek-R1` shows a more gradual ascent. Per-step error (right) further highlights the adaptability of SEA, as it consistently uncovers challenging inputs. The slope of per-step error varies across models: `gpt-4o-mini` shows a relatively flat trajectory, while `o1-mini` and `DeepSeek-V3` show steeper climbs.

**Ablation studies.** SEA has two significant procedures: (1) collecting and updating the source error set (line 9 in Alg. 1) and (2) directed graph construction and source pruning (lines 10 and 11 in Alg. 1). Procedure Two relies on the result from Procedure One. To analyze the contribution to both procedures, we conduct two ablation experiments on four selected LLMs (`Qwen2.5-72B-Instruct`, `gpt-4o`, `Llama-3.3-70B-Instruct`, and `DeepSeek-V3`) by removing procedure one and both procedures to create two variants of SEA: (i) w/o pruning and (ii) random selection. For the random selection, we change all error-related batches into random batches where we randomly select paragraphs from the knowledge base. We summarize the results in Fig. 4, from which we observe that the contribution of each procedure to SEA is linearly increasing. The cumulative error of random selection barely increases, while the gap between SEA and w/o pruning variant starts increasing after

| Model Cost | DeepSeek-R1 | R1-Distill-Llama-70B | o1-mini | DeepSeek-V3 | Llama-3.3-70B | Qwen2.5-72B | gpt-4o-mini | gpt-4o |
|---|---|---|---|---|---|---|---|---|
| Generation Cost (US $) | 28.163 | 28.660 | 31.094 | 30.208 | 29.776 | 29.542 | 28.243 | 32.897 |
| Inference Cost (US $) | 48.360 | 7.888 | 39.708 | 1.261 | 0.868 | 0.37 | 0.347 | 7.905 |
| Inference Output Tokens | 19,608,736 | 10,836,882 | 8,507,015 | 380,099 | 1,024,942 | 272,566 | 125,117 | 308,145 |

**Table 1:** Question generation cost, inference cost, and output tokens at inference time across 20 steps (results in Fig.3; 20,000 questions in total). We can see a significant gap between reasoning models (`DeepSeek-R1`, `R1-Distill-Llama-70B`, and `o1-mini`) and other non-reasoning models.

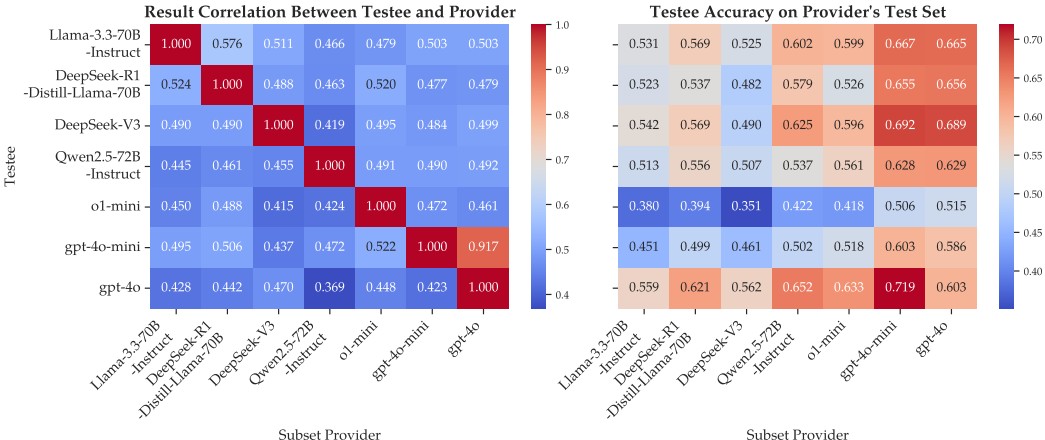

**Figure 5:** Comparison of cross-validation between each model. X-axis indicates the subset provider (i.e., $\hat{\mathcal{S}}$ provider; sourced from experiments in Fig. 3), and Y-axis denotes the testee. We summarize two results: (1) correlation between testee result and provider result, and (2) accuracy of testee on each provider's results. The higher the correlation, the more similar the answers of the two models are. Similarly, the higher the testee's accuracy, the more challenging the provider's question.

a few steps, indicating that low-quality sources that haven't been pruned by the cumulative error start negatively affecting SEA.

**Cost Analysis** We summarize all the model's costs for Fig. 3 results in Tab. 1. We use cloud API served by DeepSeek and DeepInfra for all open-sourced models (`DeepSeek-R1`, `R1-Distill-Llama-70B`, `DeepSeek-V3`, `Llama-3.3-70B`, and `Qwen2.5-72B`), and our cost calculation is based on their token price per million tokens. We observe that the variance for the generation cost is low, while that for the inference cost is high. A significant difference can be discovered by looking into the inference cost between reasoning models (`DeepSeek-R1`, `R1-Distill-Llama-70B`, and `o1-mini`) and other non-reasoning models. `DeepSeek-R1` has extremely long inference token length even for the multiple choice questions, which causes the highest cost, though its price-per-token is lower than `o1-mini` and `gpt-4o`.

## 6 Analyzing LLMs from the Discovered Deficiencies

**Query 1: How does the model perform on other LLMs' optimal subset?** In this study, we search for eight LLMs' deficiencies and consequently create eight unique optimal subsets according to Eq. 1. We conduct the cross-validation by testing each model (testee) on the other models' (providers) subsets, with correlation and accuracy results summarized in Fig. 5[2]. Our analysis revealed an asymmetrical correlation pattern, where the direction of testing significantly influenced outcomes—for instance, `gpt-4o-mini` as the testee exhibited a high correlation (0.917) with `gpt-4o` as the provider, yet the reverse yielded a notably lower correlation (0.423), suggesting that `gpt-4o-mini` may be a distilled variant optimized to emulate `gpt-4o`'s behavior, while its subset highlights divergent behaviors. Models from similar families, such as `gpt-4o` and `gpt-4o-mini`, demonstrated higher correlations, indicative of shared output patterns, whereas cross-family comparisons, like

---

[2]We omit the DeepSeek-R1 due to the budget limitation.

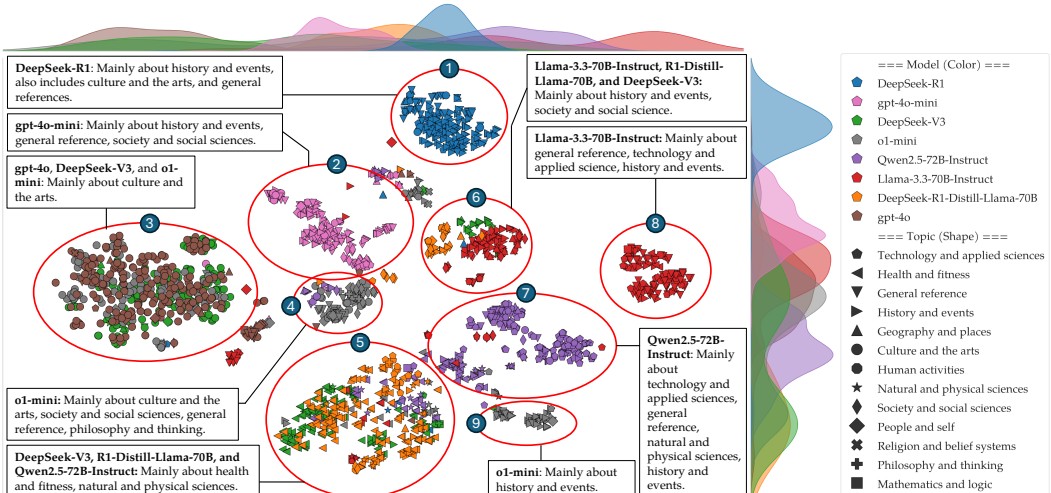

**Figure 6:** Error distribution of each testee model. We search with the **same initial set** from thirteen categories of Wikipedia. We visualize the results by t-SNE without a clustering algorithm. Each point in the figure denotes the corresponding model's source error $p \in \mathcal{P}_{\text{source}}$. Different colors denote different models, and different markers denote different categories. We can observe natural clusters of each model discovered by SEA according to their knowledge omission areas.

| Cluster ID | Models | Main Categories | Error Pattern |
|---|---|---|---|
| 3 | gpt-4o
DeepSeek-V3
o1-mini | Culture and the arts | (1) Challenging in Chronological Analysis
(2) Unfamiliar with Locational Details
(3) Issues in Pattern Recognition
(4) Inaccurate Data Synthesis
(5) Collaborative and Relational Patterns |
| 5 | Qwen2.5-72B-Instruct
Llama-3.3-70B-Instruct
R1-Distill-Llama-70B | Health and fitness
Natural and physics science | (1) Challenges with Chronological and Historical Data
(2) Issues with Contextual and Performance-Related Information
(3) Inaccurate Interpretation of Patterns and Trends
(4) Over-reliance on Assumptions and Generalizations
(5) Difficulty with Contextual Associations and Identifications |

**Table 2:** Error patterns for models in cluster 3 and 5 in Fig. 6. We aggregate error patterns from the question level to the paragraph level and finally to the model level.

Llama-3.3-70B versus gpt-4o, showed moderate correlations (0.4–0.5). The accuracy matrix illustrates model performance across subsets generated by different providers, with gpt-4o and gpt-4o-mini's subsets being less challenging, yielding higher accuracies from testee models, while DeepSeek-V3's subset posed a greater difficulty, and o1-mini consistently underperformed, underscoring varying model capabilities and subset complexities.

**Query 2: What kind of knowledge do the models lack?** To investigate this, we visualize the source error $p \in \mathcal{P}_{\text{source}}$ in $\hat{\mathcal{S}}$ of each model by compressing the embedding of $\mathcal{P}_{\text{source}}$ with t-SNE (Van der Maaten & Hinton, 2008) in Fig. 6. We mark different models in different colors, and different Wikipedia categories in different marker shapes. We can see that gpt-4o, DeepSeek-V3, and o1-mini overlap highly, with errors concentrated in culture and the arts. gpt-4o-mini and DeepSeek-R1 have unique clusters, less overlapping with other models, while both weaknesses include history and events, and society and social science. We also notice that DeepSeek-V3, R1-Distill-Llama-70B, and Qwen2.5-72B-Instruct have error overlap on health and fitness, and natural and physical sciences, while Qwen2.5-72B-Instruct also has its only cluster on technology and applied sciences, general reference, natural and physical sciences, and history and events. Errors from Llama-3.3-70B-Instruct have overlap with DeepSeek-R1-Distill-Llama-70B on history and events, indicating the possible inherent relationship between these two models. We observe that a wide range (5 out of 13) of categories can trigger o1-mini's error, including culture and the arts, society and social sciences, general reference, philosophy and thinking, and history and events, indicating a significant gap of training data and training strategies between o1-mini and other models.

**Query 3: What pattern causes a model to fail in a specific category?** To understand the behavior of a model's error, we aggregate the question-level error into paragraph-level errors and further summarize them as the model's error behavior. We choose clusters 3 and 5 in Fig. 6 and perform the multi-level aggregation by prompting an LLM to retrieve question-level error pattern, paragraph-level pattern, and finally the model and cluster-level error pattern. We only analyze clusters 3 and 5 for budget reasons and summarize the error patterns in Tab. 2. The models in cluster 3 appear to have difficulties with tasks requiring historical context, spatial awareness, and relational reasoning in the domain of culture and arts. The models in cluster 5 seem to have broader issues with contextual understanding and precision, particularly in domains requiring empirical rigor. Both clusters exhibit challenges with chronological analysis and pattern recognition, indicating that these might be common limitations across various LLMs when dealing with complex domains.

## 7 Conclusion

In this work, we introduced stochastic error ascent (SEA) that can discover knowledge deficiencies of language models on a massive knowledge base. SEA identify knowledge deficiencies in closed-weight LLMs by framing it as a budget-constrained stochastic optimization process. SEA surpass previous baselines, including ACD and AutoBencher, by uncovering 40.72 times and 26.7% more errors, respectively, at 599 and 9 times lower cost per error. SEA achieves a 100% human pass rate on generated questions, exposes distinct error clusters across models such as `gpt-4o`, `DeepSeek-V3`, and `o1-mini`, and delivers critical insights for enhancing model reliability.

## 8 Future works

**Generalizing to other modalities.** In this work, we discuss the possibility of searching an LLM's knowledge deficiencies on a massive knowledge base by SEA. However, we did not extend it to the multimodal domain, such as images and videos. The difficulty of high-quality question-answer pair synthesis from the multimodal domain limits the extension because SEA requires a high-quality question as the base for deficiency searching. Zhang et al. (2024a) suggests a possible solution for generating a benchmark from a massive image base, including 2D and 3D scenes, but they leverage human-level annotation for each image. Zhang et al. (2024b) further adopts low-level task-specific models for automatic image annotation, while the quality of the generated annotation is low. Such low-quality annotation further affects the quality of the question. Given the low quality of questions synthesized from images, it is hard for SEA to generalize to the image domain. Future work may focus on generating high-quality annotations from an image, enabling noise-reduced evaluation by SEA.

**Enlarge the searching scope.** SEA can search LLMs' vulnerabilities across a massive knowledge base, but this result is affected by the initial set. Although we demonstrated that SEA works well on different initial sets (as shown in Fig. 6 and Fig. 7) under conditioning and fully random settings, its searching scope is still limited by the high cost of close-weight LLMs. We discuss a possible solution for extending the searching scope by fitting a small model to the model's existing error in Appendix A, but it cannot fit the model's failure pattern well, resulting in less than 70% of accuracy on the test set. Future work may focus on how to enlarge the scope of the search by effectively capturing the failure pattern from the input model.

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

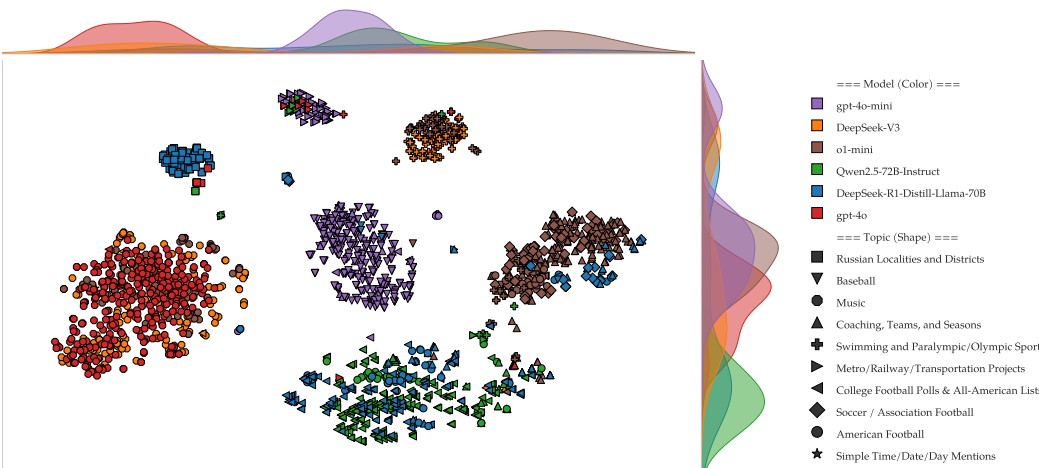

**Figure 7:** Error distribution of each testee model. We search with the **same random initial set** from Wikipedia **without specifying specific topics**. We visualize the results by t-SNE without a clustering algorithm. Each point in the figure denotes the corresponding model's source error $p \in \mathcal{P}_{\text{source}}$. Different colors denote different models, and different markers denote different categories.

## A  Extra Analysis and Case Studies

**Query 4: Will LLM produce misinformation on its unknown knowledge?**  In order to investigate this question, we randomly sampled 5 questions from the `gpt-4o` optimal subset, specifically selecting those where SEA previously identified factual errors or vulnerabilities in LLMs, representing unknown knowledge. In this experiment, we modify these questions into a free-response format using `gpt-4o` and retest them on `gpt-4o`. Errors such as Tab. 3 (incorrect doctoral year), Tab. 4 (misidentified event), Tab. 5 (wrong attribution of artist and medium), Tab. 6 (misattributed venue), and Tab. 7 (erroneous exhibition location) clearly show that when LLM encounter unknown knowledge, they may produce misinformation, as highlighted by the underlined errors in each example.

**Query 5: Will LLM exhibit memory-context conflict when answering questions related to the detected deficiencies?**  Memory-context conflicts are known as conflicts between pre-trained parametric knowledge and retrieved information (Su et al., 2024; Zhao et al., 2025). To evaluate whether LLM exhibits such a memory-context conflict when addressing questions probing their deficiencies, we randomly sampled 1000 incorrect questions from the `gpt-4o` QA set and augmented each with its corresponding retrieved factual context, testing on `gpt-4o`. Despite this external supplementation, the accuracy of `gpt-4o` improved only to 28.6%, indicating that the model adopts the provided context in merely about one-third of cases while predominantly relying on its pre-trained internal knowledge in the remaining instances. This outcome indicates that even essential external-augmented information may not sufficiently override LLM's entrenched memory. Therefore, our findings mean that LLM do exhibit a clear memory-context conflict. The relevant testee prompt is listed in Tab. 12.

**Query 6: What deficiencies can be discovered from a random initial set without category constraints?**  As described in Sec. 4, our random initial batch is uniformly sampled across 13 Wikipedia categories. However, the downstream task may lack well-defined categories. To investigate the solution, we test six models (`gpt-4o`, `gpt-4o-mini`, `o1-mini`, `Qwen2.5-72B-Instruct`, `DeepSeek-R1-Distill-Llama-70B`, and `DeepSeek-V3`) by adopting a complete random initial batch without any topic constraint when performing SEA. We search each model for 20 steps with the same setting as described in Sec. 4, summarizing the topic from each model by LDA (Blei et al., 2003) and aggregating the topic from all models into 10 general topics, including: `Baseball, American Football, Metro/Railway/Transportation Projects, Swimming/Paralympic/Olympic Sports, Music, Soccer/Association Football, Simple Time/Date/Day Mentions, College Football`

---

**Example 1**

- - - - - - - - - - - - - - - - - - - - - - - - - - - - - - - - - - - - - - - - - - - - - - - - - - -

**Title:**
James B. Stump/Career

**Original question:**
What year did James B. Stump receive his doctoral degree from Boston University?

**Ground true answer:**
2000

**Modified question:**
Discuss the year in which James B. Stump was awarded his doctoral degree from Boston University.

**Misinformation:**
James B. Stump was awarded his doctoral degree from Boston University in **1998**.

---

**Table 3:** Example 1 for query 4. The correct doctoral year is "2000", but the misinformation incorrectly states "1998". The incorrect information has been highlighted using underlines.

`Polls/All-American Lists`, `Coaching/Teams/Seasons`, and `Russian Localities/Districts`. These topics are mainly about sports and health, identifying systematic failure patterns of different LLMs in this area. We further visualize the result of the source error in Fig. 7. We first observe a similar distribution as in Fig. 6, where `gpt-4o`, `DeepSeek-V3`, and `o1-mini` share similar failure patterns, while `Qwen2.5-72B-Instruct` and `DeepSeek-R1-Distill-Llama-70B` share similar failure patterns. We notice a large volume of `DeepSeek-R1-Distill-Llama-70B` and `o1-mini` aligns with the observation in Fig. 6. We also observe that `gpt-4o`, `DeepSeek-V3`, and `o1-mini` mainly fail in music-related paragraphs, while `gpt-4o-mini` mainly fails in Baseball and Transportation project-related topics.

**Query 7: How can we extend the searching scope?** Following the settings in Zhang et al. (2024a), we try fitting a BERT (Devlin et al., 2019) model to identify if a paragraph from a knowledge base can trigger an LLM's error. We first collect 4,402 retrieved paragraphs from 50 rounds of SEA searching process on `gpt-4o`. We annotate the paragraphs as 0 if the average accuracy across the generated questions is lower than 0.5, and 1 otherwise. The collected paragraphs are split into training, validation, and test sets, respectively, with the ratio 8:1:1. We adopt early-stopping to prevent overfitting according to the validation performance. We tried the `bert-base-uncased` and `bert-large-uncased` respectively. The `bert-base-uncased` achieves 66.22% average accuracy on the test set, while `bert-base-uncased` achieves 67.85% average accuracy. These results suggest that larger BERT models can capture the subtle semantic cues that differentiate paragraphs likely to mislead an LLM. However, the overall performance indicates that this is a challenging classification task, potentially due to the noisy or indirect relationship between paragraph content and downstream model behavior.

## B  Prompt of SEA

This section is supplemented with some additional details when implementing the pipeline of SEA, which is introduced in Section 3.1. To be more specific, Tab 8, Tab 9, Tab 10, Tab 11, and Tab 12 are prompts for multiple choice question generation, question rephrasing, analyzing error pattern, SEA testee model, and testee model in query 5, respectively.

**Example 2**

- - - - - - - - - - - - - - - - - - - - - - - - - - - - - - - - - - - - - - - - - - -

**Title:**
Mimi Cherono Ng'ok/Group exhibitions

**Original question:**
In which event was the series "Do You Miss Me? Sometimes, Not Always" by Mimi Cherono Ng'ok included?

**Ground true answer:**
African Photography Encounters

**Modified question:**
Discuss the event in which Mimi Cherono Ng'ok's series "Do You Miss Me? Sometimes, Not Always" was included. Provide details about the event and its significance.

**Misinformation:**
Mimi Cherono Ng'ok's photographic series "Do You Miss Me? Sometimes, Not Always" was included in **the 2019 Whitney Biennial**, a prestigious contemporary art exhibition held at the Whitney Museum of American Art in New York City. The Whitney Biennial is one of the most significant events in the art world, known for showcasing the latest trends and emerging voices in American art. It serves as a barometer of contemporary art in the United States, often highlighting diverse perspectives and innovative practices.The 2019 edition of the Biennial was particularly noted for its focus on issues of identity, social justice, and the political climate, reflecting the broader cultural and social dynamics of the time. Mimi Cherono Ng'ok's work, which often explores themes of memory, displacement, and emotional landscapes, fit well within this context. Her series "Do You Miss Me? Sometimes, Not Always" delves into personal and collective narratives, capturing the nuances of absence and longing through evocative imagery.The inclusion of Ng'ok's work in the Whitney Biennial not only elevated her profile as an artist but also contributed to the broader discourse on contemporary African art and its place within the global art scene. Her participation underscored the Biennial's commitment to presenting diverse artistic voices and addressing complex themes that resonate with audiences worldwide.

**Table 4:** Example 2 for query 4. The proper event is "African Photography Encounters," yet the misinformation erroneously identifies it as the "2019 Whitney Biennial". The incorrect information has been highlighted using underlines.

**Example 3**

------------------------------------------------------------------------

**Title:**
Museo Nacional Centro de Arte Reina Sofía/Gallery

**Original question:**
Identify the artist and medium for the 1923 painting 'Portrait of Tristan Tzara'.

**Ground true answer:**
Robert Delaunay, oil on cardboard

**Modified question:**
Describe the artist and the medium used for the painting 'Portrait of Tristan Tzara' created in 1923.

**Misinformation:**
The painting 'Portrait of Tristan Tzara' was created by the Romanian avant-garde artist **Marcel Janco** in 1923. Marcel Janco was a key figure in the Dada movement, which was known for its anti-establishment and anti-art sentiments. The medium used for this painting is **oil on canvas**. Janco's work often incorporated elements of abstraction and cubism, reflecting the experimental and revolutionary spirit of the Dada movement. Tristan Tzara, the subject of the portrait, was a prominent Dada poet and one of the movement's founders, making this painting a significant piece in the context of early 20th-century avant-garde art.

**Table 5:** Example 3 for query 4. It shows that the true artist and medium are "Robert Delaunay, oil on cardboard", while the misinformation wrongly lists "Marcel Janco" and "oil on canvas". The incorrect information has been highlighted using underlines.

**Example 4**

- - - - - - - - - - - - - - - - - - - - - - - - - - - - - - - - - - - - - - - - - - - -

**Title:**
Paa Joe/Exhibitions/Solo and group exhibitions

**Original question:**
Identify the venue that hosted the "Six Feet Under" exhibition during the years 2007 and 2008.

**Ground true answer:**
Deutsches Hygiene-Museum, Dresden

**Modified question:**
Describe the venue that hosted the "Six Feet Under" exhibition during the years 2007 and 2008.

**Misinformation:**
The "Six Feet Under" exhibition was hosted at the **Kunstmuseum Bern in Switzerland** during the years 2007 and 2008. The Kunstmuseum Bern is renowned for its extensive collection of fine art, ranging from the Middle Ages to contemporary works. The museum is situated in the Swiss capital, Bern, and is known for its modern architectural design that complements its historical setting. The "Six Feet Under" exhibition explored themes related to death and the afterlife, featuring a diverse array of artworks that engaged with these profound topics.

**Table 6:** Example 4 for query 4. In Example 4, the accurate venue is "Deutsches Hygiene-Museum, Dresden", but the misinformation mistakenly mentions "Kunstmuseum Bern". The incorrect information has been highlighted using underlines.

**Example 5**

- - - - - - - - - - - - - - - - - - - - - - - - - - - - - - - - - - - - - - - - - - - - - - - -

**Title:**
Clare Kenny/Selected exhibitions

**Original question:**
Which art space featured Clare Kenny's "If I was a Rich Girl" in 2019?

**Ground true answer:**
Kunst Raum Riehen

**Modified question:**
Identify the art space that showcased Clare Kenny's exhibition titled "If I was a Rich Girl" in the year 2019. Provide details about the venue and its significance in the art community if possible.

**Misinformation:**
Clare Kenny's exhibition titled "If I was a Rich Girl" was showcased at **VITRINE** in 2019. VITRINE is a contemporary art gallery located in London, known for its unique exhibition format that utilizes both a window gallery space and an interior gallery. This allows for art to be viewed from the street, making it accessible to a broader audience. VITRINE has gained a reputation for supporting emerging and mid-career artists, providing them with a platform to present innovative and experimental works. The gallery's commitment to showcasing diverse and thought-provoking art has made it a significant player in the contemporary art scene.

**Table 7:** Example 5 for query 4. Original testing process correctly names the venue as "Kunst Raum Riehen", in contrast to the misinformation's incorrect attribution to "VITRINE".The incorrect information has been highlighted using underlines.

**Prompt for Multiple Choice Question-answer Pair Generation**

According to the following title and context, reply with {num_of_qa} multiple-choice questions and a statement based on the question.

## Title
{title}

## Context (Only you can see this context)
{context}

## Requirement
- Each question should include 4 options.
- The context is only for you to generate the question. Therefore, your question should not redirect to this context, like "according to the provided context...".
- Your question should include enough information that can help the testee to understand the background.
- Your reply should follow the given JSON format.

## Reply format

```
[
    {
        "question": "QUESTION CONTENT",
        "options": ["A: ...", "B: ...", "C: ...", "D: ..."],
        "statement": "STATEMENT OF THIS QUESTION.",
        "answer": "CHOICE FROM THE OPTIONS. For example, A"
    },
    ...
]
```

**Table 8:** The prompt for multiple choice question generation.

**Prompt for Question Rephrasing**

According to the following title, context, and question, reply with {num_of_qa} rephrased questions and the corresponding answers. Your question should provide sufficient content to avoid ambiguity. You should reply with JSON format as follows:

## Title
{title}

## Context (Only you can see this context)
{context}

## Question
{question}

## Requirement
- Your question should have the same meaning as the provided question, only rephrased.
- The context is only for you to generate the question. Therefore, your question should not redirect to this context, like "according to the provided context...".
- Your question should include enough information that can help the testee to understand the background.
- Your reply should follow the given JSON format.

## Reply format

```
[
    {
        "question": "QUESTION CONTENT",
        "options": ["A: ...", "B: ...", "C: ...", "D: ..."],
        "statement": "STATEMENT OF THIS QUESTION.",
        "answer": "CHOICE FROM THE OPTIONS. For example, A"
    },
    ...
]
```

**Table 9:** The prompt for Question Rephrasing.

**Prompt for Analyzing Error Pattern**

Given a context, a question, and its corresponding incorrect solution, generate a gerund phrase that thoroughly and precisely describes the **specific** skill or capability lacking that causes the error.

## Context
{context}

## Question
{question}

## Correct Solution
{answer}

## Incorrect Solution
{llm_answer}

## Requirement
- The incorrect Solution is provided by a testee who cannot access the context. Your answer should not mention that the skill is related to context information retrieval.
- The skill description should be an action-oriented gerund phrase that is **informative** and **detailed**.
- The phrase should refer to a **specific** skill or capability that comprehensively covers the key aspects of the solution, without including any context or specifics from the question or solution.
- Avoid unnecessary elements unrelated to the core capability.
- Please output **only a gerund phrase** describing the skill, with NO additional text.

Table 10: The prompt for Analyzing Error Pattern.

**Prompt for Testee Model**

Given the topic: {topic}, answer the following question by choosing one option in Options:

Question:
{que}

Options:
{opts}

Your Answer (put your answer in \box{}):

Table 11: The prompt for Testee Model.

**Prompt for Testee Model in Query 5**

Given the topic: {topic}, answer the following question by choosing one option from Options below:

Question:
{question}

Retrieved Fact:
{input}

Options:
{options}

Answer:

**Table 12:** The prompt for Testee Model in Query 5.

