# OpenReview forum: "Discovering Knowledge Deficiencies of Language Models on Massive Knowledge Base"
_colmweb.org/COLM/2025/Conference — COLM 2025_

### Official Review · Reviewer_UD1R · 2025-04-27

**Rating:** 6
**Confidence:** 4
**Ethics Flag:** 1

**Summary:**

The authors propose stochastic error ascent (SEA), an efficient framework for discovering knowledge deficiencies in closed-weight LLMs under a query budget. SEA iteratively retrieves new high-error candidates by leveraging the semantic similarity to previously observed failures, and employs hierarchical retrieval across documents driven by a relation directed acyclic graph-based error propagation method. SEA discovers knowledge deficiencies of various LLMs, showcasing their diverse vulnerabilities.

**Questions To Authors:**

Detailed Questions:
1. I am wondering, if you randomly select your first batch of passages, will the variance of your method be very high? I didn't see any analysis about the influence of starting SEA from different initial passage groups. Can you tell me more about it?
2. Why $\xi=\gamma$? At each step t, you will not include any new passage whose error rate is $< \xi$, meaning that you won't prune any passage already in your source set, right? Also, if you compute the cumulative error for a source passage's descendants, when a new passage is include into the source set, it has no descendant, then how do you deal with them? I think this part is rather unclear and confusing. I hope more discussion can be added. Maybe you could provide an example in the rebuttal.
3. I wish to see more analysis about various values of $\xi$ and $\gamma$. I think it may be some variant changing the observation.
4. I don't know what you mean by "we set the budget as 20,000 API calls for fclose plus the cost for QA generation". This doesn't seem a clear description of budget. How much will it cost for QA generation in SEA and ACD? What if each API call asks for different amount of information?
5. Similarly, what do you mean by "and the budget as AutoBencher to generate 13 benchmarks when comparing with AutoBencher." Can you quantify that? Since the efficiency is one of your main claimed contribution. I think it is important.
6. What version of wikipedia are you using? Knowledge might be outdated if you use previous versions of knowledge base. It might have a conflict with models' parametric knowledge.

**Reasons To Accept:**

1. Interesting topic of dynamically detect LLMs' failure with an evolving evaluation set.
2. Well-written paper and extensive experiments. I also like the figures.
3. The proposed method is useful and it discovers interesting findings in understanding various models.

**Reasons To Reject:**

Some technical details are not clear. Please see "Questions To Authors".
I will also carefully read other reviewers' comments and authors' rebuttal to have a more accurate judgment.

---

> ### Author Response · Authors · 2025-05-30
> **Author response to Reviewer UD1R (Part 1/2)**
>
> We sincerely thank the reviewer for their detailed and insightful questions. This feedback is invaluable for improving the clarity of our work. We provide the following responses to address each point.
>
> > **1. On the Influence of the Initial Passage Group**
>
> We recognize that the initial random batch could influence the search trajectory. To address this, we performed a specific experiment to measure the sensitivity of SEA to the initial seed selection.
> In our main experiments, the initial batch $B$ was created by uniformly sampling paragraphs from 13 predefined Wikipedia categories to ensure a broad starting point.
> However, in `Appendix A (Query 6)`, we detail an additional experiment where we initialized SEA with a **completely random batch of paragraphs from Wikipedia, with no category constraints at all**.
> The results of this experiment are visualized in `Figure 7`. We found that the resulting error distributions were highly similar to those from our main experiment (`Figure 6`). For example, the same model clusters (e.g., `gpt-4o`, `DeepSeek-V3`, and `o1-mini` failing on similar topics) emerged independently of the starting point.
> This demonstrates that while the initial batch seeds the search, the iterative, error-driven nature of SEA is robust and consistently converges on a model's inherent knowledge deficiencies rather than being highly variable based on the start.
>
> > **2. On Thresholds ($\xi$ and $\gamma$) and DAG Logic**
>
> We appreciate the reviewer pointing out the need for greater clarity here. Let us break down the logic.
>
> **Why $\xi = \gamma$?** Setting $\xi = \gamma = 0.5$ is a design choice for simplicity, but the two thresholds serve different functions in the DAG.
> - **$\xi$ (Source Error Threshold)**: This threshold determines if a newly evaluated paragraph $p$ becomes a source error. It is based on the model's error rate for that single paragraph, $T_{\{p\}}$. If $T_{\{p\}} > \xi$, it is added to the source set $P_{\text{source}}$.
> - **$\gamma$ (Source Pruning Threshold)**: This threshold determines if an existing source error gets pruned. It is based on the **cumulative error $\pi_{\mathcal{G}(p)}$**, which is the *average error across all of its descendants in the DAG*.
>
> You are correct that a newly added source error will not be immediately pruned. When a paragraph first becomes a source error, it has no descendants, so the pruning condition $\pi_{\mathcal{G}(p)}< \gamma$ does not apply. It can only be pruned in later steps after it has generated descendants whose average error rate is low, indicating it's no longer a fruitful path for discovering new errors.
>
> **Example of the Process**: Given $\xi = \gamma = 0.5$.
> - **At step t**: SEA evaluates paragraph $p_1$. The model's error rate $T_{\{p_1\}}$ is $0.8$. Since $0.8 > \xi$, $p_1$ is added to the source set. It has no descendants yet.
> - **Step t+1**: Using $p_1$ as a source, SEA discovers and evaluates $p_2$ and $p_3$.
>     - The model's error on $p_2$ is $T_{\{p_2\}}= 0.9$ (a new source error).
>     - The model's error on $p_3$ is $T_{\{p_3\}}= 0.1$ (not a source error).
> - **Pruning Check for** $p_1$: $p_1$ now has two descendants, $p_2$ and $p_3$. Its cumulative error is $\pi_{\mathcal{G}(p_1)} = (0.9 + 0.1) / 2 = 0.5$. Since $\pi_{\mathcal{G}(p_1)}$ is not less than $\gamma$, which is $0.5$, $p_1$ is **not pruned** and remains a source for the next step.
>
> > **3. On Analyzing Various Values of $\xi$ and $\gamma$**
>
> Due to the budget limitation, we did not perform a detailed sweep of these hyperparameters. However, **our ablation studies in Section 5 (Figure 4) provide a quantitative analysis of their impact**. Specifically, the "w/o Pruning" variant is equivalent to setting the pruning threshold $\gamma$ to 0. The results show a clear performance drop compared to the full SEA framework, demonstrating that the pruning mechanism is crucial for performance. The gap between SEA and the "w/o Pruning" variant increases over time, indicating that low-quality sources that haven't been pruned by the cumulative error start negatively affecting SEA. This quantitatively validates the importance of the pruning component governed by the $\gamma$ threshold.

---

> > ### Author Response · Authors · 2025-05-30
> > **Author response to Reviewer UD1R (Part 2/2)**
> >
> > > **4. & 5. On Budget Clarity and Quantification**
> >
> > **Budget vs. ACD (Monetary Cost & API Calls)**: The phrase "we set the budget as 20,000 API calls for $f_{\text{close}}$ plus the cost for QA generation" refers to a total monetary budget cap for the entire process. The QA generation cost depends on different models when evaluated because each model’s incorrect paragraphs can be varied. You can find the cost for each model in `Appendix B (Table 2)`. We use the exact same amount of model-specific cost for ACD. As shown in `Figure 2`, for the same monetary budget, the average cost to find one error with SEA was **$0.006**, while for ACD it was **$3.595**. This represents a **599x reduction in cost** per discovered error.
> >
> > **Budget vs. AutoBencher (Problem Size)**
> > The comparison with AutoBencher was based on a fixed problem size, not a monetary budget, because AutoBencher is designed to take a topic and iteratively build a challenging benchmark dataset. Its primary output is a dataset of a specific size. In the paper's experiment, AutoBencher was configured to generate 13 benchmarks, resulting in a fixed set of 2,000 questions in total. Therefore, the most logical way to compare it with SEA is to evaluate their efficiency in this specific task. The "budget" was therefore defined as the problem size (2,000 questions). The results in `Figure 2` show that for the same number of questions, SEA consistently achieves a higher error rate across all models (an average of **0.38 for SEA vs. 0.30 for AutoBencher**), representing a **26.7% relative increase in error detection efficiency**. Furthermore, the cost-per-error for AutoBencher was $0.054, **9 times higher** than SEA's $0.006.
> >
> > > **6. On the version of Wikipedia database**
> >
> > We use the Wikipedia data provided by Wikimedia with a cutoff before `Dec. 20, 2024`. We will make it clearer in the final version.

---

> > > ### Comment · Reviewer_UD1R · 2025-05-31
> > > **Thank You for the Response**
> > >
> > > I like this paper and authors have addressed most of my concerns. I recommend exploring different value settings regarding the Source Error Threshold and the Source Pruning Threshold.
> > >
> > > I have a further question after looking at your discussion with Reviewer cTGP. I am wondering could you also somehow provide the comparison between the distribution of the errors found in SEA and of the ones found in ACD? Since SEA finds next batch of errors from existing errors and ACD finds new errors through "discarding proposals that are similar to existing tasks". Will this cause SEA to sacrifice diversity when it detects errors?
> > >
> > > I will raise my confidence for now and I wish to see further comments by the authors.

---

> > > > ### Author Response · Authors · 2025-06-01
> > > > **Author response to Reviewer UD1R about the further question**
> > > >
> > > > Thank you for your insightful question and for highlighting the comparison with ACD's approach to error discovery.
> > > >
> > > > To answer your question, we compare the categorical diversity of errors found by SEA and ACD. For each LLM, we mapped errors from both methods to the 13 top-level Wikipedia categories used in our study. We choose Shannon entropy to evaluate the diversity of topics detected by both methods: $H=−\sum_{i=1}^R ​p_i \log(p_i​)$, where $R$ is the total number of categories (13 in our setting), and $p_i$​ is the proportion of errors in category $i$. A higher $H$ value indicates greater diversity (i.e., errors are more evenly spread across more categories). We then calculated: (1) the number of unique categories containing errors, and (2) the Shannon diversity index across these categories. The final results of entropy are shown below:
> > > >
> > > > |     | **DeepSeek-R1** | **DeepSeek-R1-Distill-Llama-70B** | **DeepSeek-V3** | **Llama-3.3-70B-Instruct** | **Qwen2.5-72B-Instruct** | **gpt-4o** | **gpt-4o-mini** | **o1-mini**  |
> > > > |-----|-----------------|-----------------------------------|-----------------|----------------------------|--------------------------|------------|-----------------|--------------|
> > > > | ACD | 0.2831          | 0.2206                            | 0.1842          | 0.3042                     | 0.2484                   | 0.0992     | 0.2600          | (no results) |
> > > > | SEA | 0.6552          | 0.7447                            | 0.9781          | 1.0105                     | 1.0125                   | 0.6424     | 0.8273          | 1.2839       |
> > > >
> > > > As the table demonstrates, SEA consistently achieves a higher diversity index across all compared models. For instance, with `DeepSeek-V3`, SEA achieved a diversity index of `0.9781`, whereas ACD's was `0.1842`. Similarly, for `Llama-3.3-70B-Instruct`, SEA's index was `1.0105` compared to ACD's `0.3042`.
> > > >
> > > > These results strongly suggest that SEA does not sacrifice, and in fact significantly enhances, categorical error diversity compared to ACD. While SEA leverages semantic similarity to previously observed failures to efficiently find new error candidates, its hierarchical retrieval across document and paragraph levels, combined with starting from a diverse initial set drawn from multiple Wikipedia categories, allows it to explore a broader range of knowledge areas. ACD's approach, while aiming for novelty by discarding similar proposals, uncovers far fewer errors overall (as shown in our paper, `Figure 2`), which inherently limits the diversity of error categories it can identify and report within practical evaluation budgets.

---

> > > > > ### Comment · Reviewer_UD1R · 2025-06-05
> > > > > **Further Response from Reviewer**
> > > > >
> > > > > Ok. I get it. It seems the diversity largely comes from how you choose these 13 categories, right?

---

> > > > > > ### Author Response · Authors · 2025-06-05
> > > > > > **Author Response to reviewer UD1R**
> > > > > >
> > > > > > Thank you for your reply. While the 13 categories defined by Wikipedia provide a broad starting point, SEA’s high diversity does not depend on this alone. SEA actively searches for new high-error candidates and naturally shifts focus when one error type is saturated, guided by the diversity of prior failures. Notably, `Figure 7 (Appendix)` shows SEA uncovers diverse error clusters even when initialized with random paragraphs, demonstrating its inherent ability to identify varied knowledge gaps. The broad initialization helps, but SEA’s core mechanisms drive diverse and adaptive error discovery.

---

### Official Review · Reviewer_cJFu · 2025-05-13

**Rating:** 6
**Confidence:** 4
**Ethics Flag:** 1

**Summary:**

The paper "Discovering Knowledge Deficiencies of Language Models on Massive Knowledge Base" introduces Stochastic Error Ascent (SEA), a novel framework for uncovering factual weaknesses in large language models (LLMs), particularly those with closed weights. Existing benchmarks for evaluating LLM factuality are limited in scale and static in scope. SEA addresses this by formulating error discovery as a stochastic optimization problem under a strict query budget. Instead of exhaustive probing, SEA iteratively identifies error-prone knowledge regions using semantic similarity to past model failures. It employs hierarchical document-paragraph retrieval and a relation-directed acyclic graph (DAG) to model error propagation and prune low-yield sources, maximizing discovery efficiency.

Empirical results show that SEA discovers 40.7× more errors than Automated Capability Discovery and 26.7% more than AutoBencher, while drastically reducing the cost per error. Human evaluation confirms the high validity of SEA-generated questions. Further analyses reveal consistent intra-family failure patterns and overlapping model-specific deficiencies, especially in domains like history, culture, and science. SEA's adaptive querying and systematic diagnosis highlight its potential to guide future LLM development via targeted fine-tuning and enhanced data coverage. This work establishes SEA as a scalable, efficient alternative for uncovering hidden vulnerabilities in LLMs.

**Reasons To Accept:**

1. Novel and Scalable Framework
The proposed Stochastic Error Ascent (SEA) is a novel contribution that frames knowledge deficiency discovery in LLMs as a budget-constrained stochastic optimization problem, a formulation not previously explored in this domain. SEA’s approach enables efficient, scalable probing of massive knowledge bases without requiring full model access or exhaustive sampling.

2. Empirical Superiority Over Baselines
SEA demonstrates substantial improvements over existing methods:

40.7× more errors discovered than Automated Capability Discovery (ACD).
26.7% more than AutoBencher.

**Reasons To Reject:**

Dependence on Black-Box Models and External Generators
SEA relies heavily on an external LLM (e.g., GPT-4o) for question generation and evaluation. This introduces a dependency that can obscure the true deficiencies of the tested models, especially if the generator itself has biases or blind spots. It raises concerns about evaluation leakage and the circular use of LLMs to evaluate LLMs.

---

> ### Author Response · Authors · 2025-05-30
> **Author response to Reviewer cJFu**
>
> We thank and agree with the reviewer for the issue that the potential for circular evaluation and bias is a critical issue in LLM research. In our paper, we have shown our efforts into mitigating these risks. We also make further clarification in the following:
>
> > **Clarifying the Role of the Generator and Ensuring Quality**
>
> We want to clarify that the external LLM (`gpt-4o`) in our framework does not act as a subjective "judge" but as a structured data-to-question generation tool. Its role is to convert a factual paragraph from the Wikipedia knowledge base into a multiple-choice question where the answer is explicitly contained within the provided text. The final evaluation of the model being tested is not based on another LLM's opinion, but on an objective comparison against the ground truth from the source document. To address the concern about the generator's own biases or blind spots, we implemented a rigorous human validation process.
>
> **Human Evaluation of Generated Questions**: We thoroughly evaluated the questions created by our `gpt-4o` generator. Five college-level students were tasked with verifying the generated questions and answers by cross-referencing them with the original source paragraphs from the knowledge base.
>
> **100% Pass Rate**: Out of 1,000 randomly sampled questions generated by SEA, this evaluation achieved a **100% human pass rate**. This result confirms that the generated questions are factually grounded and that their correct answers are verifiably present in the source text.
>
> **Objective Evaluation**: The final accuracy is computed by simply comparing the testee model's chosen option with the ground truth answer derived directly from the source paragraph. This deterministic process removes the subjectivity and potential leakage associated with using an LLM for qualitative evaluation.
>
> This methodology of using a powerful LLM for structured content generation from source documents has also been adopted by other recent work in the field, such as AutoBencher. Our robust validation process ensures that our framework reliably identifies the true knowledge deficiencies of the tested models rather than being an artifact of the generator.

---

> ### Author Response · Authors · 2025-06-07
> **Thanks for your effort!**
>
> Dear Reviewer cJFu,
>
> We sincerely thank you for your detailed and thoughtful review of our paper. We have done our best to address the concerns you raised, especially regarding the dependence on an external generator and your other valuable points. Your feedback is crucial for improving our work.
>
> As the discussion deadline is approaching, we wanted to kindly check if our responses have clarified these issues. If any aspect of our explanation remains unclear, we are ready to provide further details immediately.
>
> We are highly encouraged if your concerns have been addressed, and we are on standby to offer any additional clarification needed before the deadline passes.
>
> Thank you again for your time and guidance.
>
> Best,
>
> Authors

---

### Official Review · Reviewer_PkVq · 2025-05-13

**Rating:** 7
**Confidence:** 3
**Ethics Flag:** 1

**Summary:**

This work proposes SEA, a scalable and efficient framework for discovering knowledge deficiencies (errors) in closed-weight LLMs under a strict query budget. it iteratively retrieves new high-error candidates by leveraging the semantic similarity to previously observed failures. SEA can uncover more knowledge errors from LMMs than the existing method Automated Capability Discovery and AutoBencher, while substantially reducing the cost-per-error.

**Questions To Authors:**

Providing a case analysis to visually demonstrate the objectives, process, and possible outcomes of the task.

**Reasons To Accept:**

This work presents a novel methodology for probing the capability boundaries of large language models (LLMs), with particular focus on identifying potential knowledge errors. The research demonstrates well-defined objectives and yields remarkably encouraging results.

**Reasons To Reject:**

Some details need to be further clarified:
1.	DAG Modeling Rationale: Why is a Directed Acyclic Graph (DAG) used to model the entire decision process? What principles govern the edge construction in the DAG?
2.	Initial Seed Selection: How are the initial seeds selected? Does seed selection have impact on the final results?
3.	Experimant details: How to construct the experimental dataset? What is the specific schema/structure of the experimental dataset? How do different threshold values quantitatively impact error detection rates?

---

> ### Author Response · Authors · 2025-05-30
> **Author response to Reviewer PkVq (Part 1/2)**
>
> We sincerely thank the reviewer’s thoughtful feedback and questions. We provide the following clarifications to address the requested details.
>
> > **On the Rationale for DAG Modeling**
>
> **Rationale**: The primary purpose of the relation DAG is to **identify and model systematic weaknesses** within the language model. Large language models often exhibit correlated failure patterns, where errors are not isolated but are concentrated in specific knowledge areas. The DAG structure allows us to model these relationships explicitly by tracing **error propagation paths**. By representing paragraphs that induce errors (source errors) as nodes, we can track how one error-prone topic leads to the discovery of other, semantically similar error-prone topics over time. This helps us understand if a model has a fundamental deficiency in, for example, "chronological reasoning" within the broader domain of "history."
>
> **Edge Construction**: The principles are based on semantic similarity and error propagation. An edge is constructed as follows:
> At each step $t$, we have a set of source error paragraphs, $P_{\text{source}}^{(t)}$, collected according to LLM’s incorrect answer.
> We use these source errors to find a new batch of error-inducing paragraphs in the next step, $P_{\text{source}}^{(t+1)}​$, by retrieving semantically similar candidates as in `Figure 1`.
> A **directed edge** is drawn from a paragraph $p \in P_{\text{source}}^{(t)}​$ to its semantically similar descendants in $P_{\text{source}}^{(t+1)}​$ that also induce errors.
> To ensure the graph remains **acyclic**, we remove any newly discovered source errors from the knowledge base $K$, which prevents them from being discovered again and creating loops.
>
> > **On the Initial Seed Selection**
>
> **Selection Process**: For our main experiments, the initial batch $B$ was constructed by uniformly retrieving 40 paragraphs from 13 predefined top-level Wikipedia categories. This was done to ensure a broad and category-agnostic starting point for the error discovery process for all models, facilitating a fair comparison.
>
> **Impact on Results**: We investigated the sensitivity of SEA to the initial seeds. In `Appendix A (Query 6)`, we present an experiment where we initiated SEA with a **completely random initial batch, without any topic or category constraints**. The results, visualized in `Figure 7`, show that SEA discovers similar failure patterns and error distributions as in the main experiment (`Figure 6`). For example, `gpt-4o`, `DeepSeek-V3`, and `o1-mini` still form a distinct error cluster, regardless of the different starting points. This demonstrates that while the initial seed starts the search, the iterative, error-driven nature of SEA is robust and consistently converges on the model's intrinsic knowledge deficiencies.
>
> > **On Experiment Details**
>
> **Experimental Dataset and Structure**: As described in `Section 4 (Knowledge base details)`, our experiments are conducted on a massive knowledge base collected from **English Wikipedia, comprising 7.1M documents and 28.8M paragraphs**. SEA dynamically samples from this knowledge base. The "dataset" for evaluation is generated on-the-fly. For each paragraph $p$ selected by SEA, we use a generator LLM (`gpt-4o`) to create multiple-choice questions.
>
> Impact of Thresholds: The thresholds $\xi$ (for identifying a source error) and $\gamma$ (for pruning the source error set) were set to 0.5 for our experiments.Due to the budget limitation, we did not perform a detailed sweep of these hyperparameters. However, **our ablation studies in `Section 5 (Figure 4)` provide a quantitative analysis of their impact**. Specifically, the "w/o Pruning" variant is equivalent to setting the pruning threshold $\gamma$ to 0. The results show a clear performance drop compared to the full SEA framework, demonstrating that the pruning mechanism is crucial for performance. The gap between SEA and the "w/o Pruning" variant increases over time, indicating that low-quality sources that haven't been pruned by the cumulative error start negatively affecting SEA. This quantitatively validates the importance of the pruning component governed by the $\gamma$ threshold.

---

> > ### Author Response · Authors · 2025-05-30
> > **Author response to Reviewer PkVq (Part 2/2)**
> >
> > > **Providing a case analysis**
> >
> > **Scenario**: We want to test the knowledge deficiencies of a powerful, closed-weight LLM (let's call it `TestLLM`). Our knowledge base is the entirety of English Wikipedia.
> >
> > **Step 1: Initial Probing and Finding a "Source Error"**
> > 1. The process begins by selecting a random batch of paragraphs from Wikipedia to get started. For example, a Wikipedia page **"Paa Joe"**, under the section for **"Solo and group exhibitions"**.
> > 2. SEA uses a generator LLM (`gpt-4o`) to read this paragraph and create a question from it. The paragraph contains the fact that Paa Joe's work was in the "Six Feet Under" exhibition at the "Deutsches Hygiene-Museum, Dresden".
> >     - **Generated Question**: "Identify the venue that hosted the 'Six Feet Under' exhibition during the years 2007 and 2008." (Options omitted)
> >     - **Ground Truth Answer**: Deutsches Hygiene-Museum, Dresden
> > 3. The `TestLLM`, lacking this specific knowledge, selects an incorrect but plausible-sounding answer: `Kunstmuseum Bern`
> > 4. Because `TestLLM` failed, the original paragraph from the "Paa Joe" article is flagged as a **"source error"**.
> >
> > **Step 2: Stochastic Error Ascent - Finding Similar Errors**
> > Now the core of SEA begins. In this paper, we claim that if the model failed on this specific art exhibition fact, it might fail on similar facts.
> > 1. **Error-Related Retrieval**: SEA searches the entire Wikipedia knowledge base for paragraphs that are semantically similar to the "Paa Joe" source error. This search might find other articles about modern art exhibitions, museum collections, or specific artists.
> > 2. **Hierarchical Search**: To do this efficiently, the search is hierarchical. It first finds similar document abstracts (e.g., other museum pages) and then searches for similar paragraphs within those documents.
> > 3. **Evaluation of New Candidates**: Let's say the search returns a paragraph about the artist **Clare Kenny**. A new question is generated and posed to `TestLLM`.
> >     - **Generated Question**: "Which art space featured Clare Kenny's 'If I was a Rich Girl' in 2019?" (Options omitted)
> >     - **Ground Truth Answer**: Kunst Raum Riehen
> > 4. **TestLLM's Incorrect Answer**: VITRINE
> > This is another failure. The "Clare Kenny" paragraph is now also marked as a source error.
> >
> > **Step 3: Building the Relation DAG and Pruning**
> > 1. **Modeling Error Propagation**: To track how one error leads to another, SEA constructs a relation Directed Acyclic Graph (DAG). An edge is drawn from the "Paa Joe" node to the "Clare Kenny" node. This suggests a potential systematic weakness in TestLLM's knowledge about specific European contemporary art venues.
> > 2. **Source Pruning**: The process iterates. In the next step, SEA will search for paragraphs similar to both source errors. Over time, the system tracks which source errors are most effective at discovering new ones. If a source error repeatedly leads to dead ends (paragraphs the model knows), its importance score decreases, and it may be "pruned" to keep the search efficient.
> >
> > This iterative process of finding an error, searching for similar content, and evaluating it continues until the predefined budget (e.g., number of API calls) is reached.

---

> ### Author Response · Authors · 2025-06-07
> **Thanks for your effort!**
>
> Dear Reviewer PkVq,
>
> Thank you for your valuable feedback and insightful questions. We appreciate your positive assessment of our work.
>
> We have provided a detailed response addressing your points regarding the DAG modeling rationale, initial seed selection, and experimental details. We have also included a case analysis to visually demonstrate the process and outcomes of our framework, as you suggested.
>
> As the discussion deadline is approaching, we wanted to gently check if our revisions and responses have addressed your concerns. Your feedback is very important to us, and we would be grateful to know if any points remain unclear. We are available to provide further clarification as needed before the deadline.
>
> Thank you again for your time and effort in reviewing our paper.
>
> Thanks,
>
> Authors

---

### Official Review · Reviewer_cTGP · 2025-05-22

**Rating:** 6
**Confidence:** 4
**Ethics Flag:** 1

**Summary:**

This paper focuses on knowledge deficiencies discovery in language models within a strict query budget and massive knowledge base. The proposed Stochastic Error Ascent (SEA) combines semantic similarity retrieval and error propagation graphs to iteratively identify model errors. The framework is evaluated on several popular open- and closed-source large language models. Compared with baselines such as Automated Capability Discovery and AutoBencher, SEA demonstrates significant improvements in both the number and efficiency of discovered knowledge errors.

**Reasons To Accept:**

+ SEA combines stochastic optimization, semantic similarity retrieval, and error graphs for automated knowledge deficiency discovery.
+ Given a knowledge base, SEA is superior in the number of discovered errors.
+ SEA requires no access to model internals, and can be readily applied to closed-source API environments.

**Reasons To Reject:**

+ The SEA framework relies on the initial batch for error detection, and its iterative, similarity-driven search may inherit biases from this initial batch. While the authors mention a uniform retrieval strategy for the initial batch, important implementation details and a deeper discussion are lacking. It remains unclear whether all models use the same initial batch in all experiments, how each model performs on the initial batch, and more importantly, how sensitive the experimental results are to the choice of initial batch.

+ The paper lacks a quantitative and systematic analysis of the diversity and the spread rate of errors discovered by SEA, making it unclear whether SEA identifies a broad range of error types or merely concentrates on a few clusters.

+ Only comparing error counts with the ACD baseline may be unfair, since ACD tends to discard proposals that are similar to existing tasks, whereas SEA searches for errors similar to previously discovered failures. In addition, the budget comparison in the paper does not take into account the computational costs of knowledge base retrieval and embedding, which may underestimate the actual overhead of SEA.

+ The explanation of some details of the method is not clear enough. The Random Sampling in Figure 1 is not mentioned in Section 3 and is only briefly described in Section 4, without explaining the purpose or necessity of doing so.

---

> ### Author Response · Authors · 2025-05-30
> **Author response to Reviewer cTGP (Part 1/2)**
>
> We thank the reviewer for their constructive feedback and valuable suggestions. We have carefully considered the comments and provide the following clarifications to address the raised concerns.
>
> > **On the Experiment Setting on Initial Batch and Sensitivity Analysis**
>
> We appreciate the reviewer's concern regarding the potential for the initial batch to introduce bias. We would like to clarify our experimental setup and provide further evidence of SEA's robustness to the initial batch selection.
>
> **Implementation Details**: For the experiments presented in the main paper (e.g., `Figures 2, 3, 5, and 6`), the same initial batch was used for all models to ensure a fair and consistent comparison. This initial batch was randomly and uniformly retrieved 40 paragraphs from 13 predefined top-level Wikipedia categories. This strategy was chosen to provide a broad, category-balanced starting point for the error discovery process.
>
> **Sensitivity to Initial Batch**: To directly address the sensitivity question, we conducted an additional experiment where SEA was initiated with a completely random set of paragraphs from Wikipedia, without any category constraints. The results of this experiment are presented in `Figure 7` and discussed in `Appendix A (Query 6)`. The analysis shows that even with a different, entirely random start, SEA discovers similar failure patterns and distributions. For instance, `gpt-4o`, `DeepSeek-V3`, and `o1-mini` still share similar failure clusters, as do `Qwen2.5-72B-Instruct` and `DeepSeek-R1-Distill-Llama-70B`. This demonstrates that SEA's iterative, similarity-based search effectively converges on a model's inherent weaknesses, regardless of the specific starting points.
>
>
> > **On the Diversity and Spread of Discovered Errors**
>
> We thank the reviewer for pointing out the need for a more systematic analysis of error diversity. We believe that `Section 6` ("Analyzing LLMs from the Discovery Results") provides a detailed, qualitative, and quantitative analysis of the diversity of errors found by SEA, and we further clarify as follows:
>
> **Quantitative and Systematic Analysis**: In `Figure 6`, we visualize the discovered source errors ($p ∈ P_{\text{source}}$) for each model using t-SNE. The visualization is color-coded by model and uses different markers for 13 distinct Wikipedia categories. This visualization clearly shows that SEA does not merely concentrate on a few clusters but identifies diverse error patterns across a wide range of topics. For example, we observe distinct clusters of errors for different models and groups of models. `gpt-4o-mini` and `DeepSeek-R1` exhibit unique error clusters, while `gpt-4o`, `DeepSeek-V3`, and `o1-mini` overlap significantly in the "culture and the arts" category.
>
> **Error Pattern Analysis**: To further analyze the types of errors, we aggregated them from the question level to the model level for specific clusters. As detailed in `Table 2`, we analyzed the error patterns for models in clusters 3 and 5. For cluster 3 (`gpt-4o`, `DeepSeek-V3`, `o1-mini`), common failure patterns included "Challenging in Chronological Analysis" and "Unfamiliar with Locational Details". For cluster 5 (`Qwen2.5-72B-Instruct`, `Llama-3.3-70B-Instruct`, etc.), failures often involved "Challenges with Chronological and Historical Data" and "Inaccurate Interpretation of Patterns and Trends". This analysis demonstrates that SEA uncovers systematic and recurring failure modes, offering deep insights into model weaknesses beyond a simple error count.
>
> > **On the Comparison with ACD and Cost Analysis**
>
> **Fairness of ACD Comparison**: The reviewer notes that SEA and ACD have different search strategies. We acknowledge this difference; it is precisely this distinction that highlights SEA's novel contribution. While ACD discards similar proposals to broaden its search, SEA intentionally leverages semantic similarity to previously observed failures to efficiently discover new errors. Our comparison focuses on the ultimate goal of both frameworks: to discover as many model deficiencies as possible within a given budget. We compare the number of "error tasks" from ACD with the number of "source errors" from SEA, as both represent identified categories of model misinformation. The results, which show SEA finding up to 40.7x more errors, demonstrate the superior efficiency of our error-ascent approach for the task of knowledge deficiency discovery.
>
> **Computational Costs for Retrieval and Embedding**: The computational cost for retrieval and embedding is negligible compared to the LLM inference cost. We use a single 4090 GPU for the sentence transformer model serving, embedding similarity calculation in SEA, and document embedding pre-processing.

---

> ### Author Response · Authors · 2025-05-30
> **Author response to Reviewer cTGP (Part 2/2)**
>
> > **On Clarifying Methodological Details (Random Sampling)**
>
> Thanks for pointing this out. As described in the implementation details in `Section 4 (Implementation details L165)`, this step occurs after the "Top-k Error Related Retrieval". At each step, SEA first retrieves the top-k semantically similar candidates from the knowledge base (we use $k=50$) for each error. From this retrieved set of $50\times |P_{\text{incorr}}|$ paragraphs, we then randomly sample a smaller batch (40 paragraphs) to form the error-related batch $E$ for evaluation in the current step. The purpose of this random sampling is twofold: (1) It introduces stochasticity into the ascent process, helping the search escape local optima and explore a wider area within the high-error region of the embedding space, and (2) it keeps the per-step evaluation cost fixed and manageable, preventing the evaluation batch size from growing uncontrollably.

---

> ### Author Response · Authors · 2025-06-07
> **Thanks for your effort!**
>
> Dear Reviewer cTGP:
>
> We sincerely appreciate your efforts in reviewing this paper! We tried our best to address your specific concerns regarding the initial batch sensitivity, the diversity of discovered errors, the comparison with the ACD baseline, and clarifications on methodological details. Your suggestions and feedback are very important to us.
>
> The discussion deadline is approaching. We wanted to check if our responses have helped clarify these points. If you need any more clarification, we are happy to provide it as soon as possible.
>
> We are highly encouraged if your concerns have been addressed.
>
> Thanks!
>
> Authors

---

> > ### Comment · Reviewer_cTGP · 2025-06-09
> >
> > Thank you for the detailed responses. The conclusion shows that whether the 40 paragraphs are sampled evenly from all 13 categories or drawn completely at random, the distribution of discovered errors converges to a roughly similar pattern. However, if the 13 categories are balanced, a random sample of 40 paragraphs will cover at least 12 categories with around a 90% probability. Thus, selecting the initial batch from only 6 or even a single category might yield more representative findings.
> > ﻿
> >
> > I still have some concerns regarding the use of the absolute count of errors as a measure of SEA’s advantage. For example, questions such as "What's the oldest bistro in Paris?" and "What's the oldest café in Paris?" might be considered different errors, yet they share limited diversity. Showing the complete dozens of errors by ACD alongside a representative subset of errors discovered by SEA might allow for a more intuitive and direct comparison. Since SEA utilizes an external knowledge base from Wikipedia (7.1M documents), it's plausible that the number of discovered errors correlates with the size of the knowledge base. It can be beneficial to perform ablation studies using smaller Wiki knowledge bases (e.g., 1%, 10%) to assess the sensitivity of SEA. If the number of errors varies considerably with the knowledge base size, clarifying its applicability and limitations in different scenarios would enhance the paper.

---

> > > ### Author Response · Authors · 2025-06-10
> > > **Author's response to the further concerns from Reviewer cTGP**
> > >
> > > Thank you for your insightful feedback and constructive suggestions. We appreciate the thorough review and the opportunity to clarify these important points. We address your concerns below.
> > >
> > > > **On the Initial Sampling Strategy and Representativeness**
> > >
> > > We appreciate your thoughtful point on the initial sampling method. We conducted a new experiment to investigate how the initial sample's breadth impacts its diversity. We measured the mean variance of the embeddings for the initial set of 40 paragraphs, sampled from a varying number of categories for the `DeepSeek-V3` model. A higher variance suggests greater diversity in the semantic space of the starting paragraphs.
> > >
> > > | **Initial Categories** | **Mean Embedding Variance** |
> > > |------------------------|-----------------------------|
> > > | 13 Categories          | 0.064841                    |
> > > | 6 Categories           | 0.064471                    |
> > > | 1 Category             | 0.040926                    |
> > >
> > > These results show that reducing the number of starting categories has a negligible impact on diversity. SEA's iterative ascent mechanism is designed to overcome its starting conditions. As shown in our convergence analysis (`Figure 3`), the algorithm effectively navigates to diverse error regions regardless of the starting point's variance.
> > >
> > > > **On Error Diversity and Comparison with ACD**
> > >
> > > To quantitatively address this, we analyzed the categorical diversity of errors discovered by both SEA and ACD using the Shannon entropy metric: $H=−\sum_{i=1}^R ​p_i \log(p_i​)$, where $R$ is the total number of categories (13 in our setting), and $p_i$​ is the proportion of errors in category $i$. A higher $H$ value indicates greater diversity (i.e., errors are more evenly spread across more categories).
> > >
> > > |     | **DeepSeek-R1** | **DeepSeek-R1-Distill-Llama-70B** | **DeepSeek-V3** | **Llama-3.3-70B-Instruct** | **Qwen2.5-72B-Instruct** | **gpt-4o** | **gpt-4o-mini** | **o1-mini**  |
> > > |-----|-----------------|-----------------------------------|-----------------|----------------------------|--------------------------|------------|-----------------|--------------|
> > > | ACD | 0.2831          | 0.2206                            | 0.1842          | 0.3042                     | 0.2484                   | 0.0992     | 0.2600          | (no results) |
> > > | SEA | 0.6552          | 0.7447                            | 0.9781          | 1.0105                     | 1.0125                   | 0.6424     | 0.8273          | 1.2839       |
> > >
> > > As shown, SEA consistently achieves a significantly higher entropy score across all models. For instance, with Qwen2.5-72B-Instruct, SEA’s entropy is 1.0125 compared to ACD’s 0.2484. This is because ACD, relying on the model's internal knowledge without an external source, discovers a very limited number of absolute errors (e.g., only 8 errors for Qwen2.5-72B-Instruct, as shown in `Figure 2`), making broad categorical coverage impossible. This evidence quantitatively refutes the concern that SEA's higher error count stems from a lack of diversity; in fact, the opposite is true.
> > >
> > > > **On Sensitivity to Knowledge Base Size**
> > >
> > > To test SEA's sensitivity to the knowledge base size, we performed the ablation study you suggested, running SEA with a random initial batch (no topic constraint) on smaller subsets of the Wikipedia knowledge base against a random search baseline for the `DeepSeek-V3` model within a fixed budget.
> > > | **Knowledge Base Size** | **Total Paragraphs (Approx.)** | **Errors Found (Random Search)** | **Errors Found (SEA)** |
> > > |-------------------------|--------------------------------|----------------------------------|------------------------|
> > > |                      1% |                        288,000 |                               46 |                    332 |
> > > |                     10% |                      2,880,000 |                               41 |                    332 |
> > > | 100% (Full)             |                     28,800,000 |                               45 |                    335 |
> > >
> > > These results lead to a crucial conclusion: The number of errors discovered by SEA is not limited by the size of the knowledge base, but rather by the model's inherent deficiencies and the query budget. The fact that SEA finds a nearly identical number of errors in the 1% subset as it does in the full knowledge base demonstrates its remarkable efficiency. It suggests that knowledge deficiencies are not uniformly distributed but are concentrated in specific semantic "pockets." SEA's error-ascent mechanism is highly effective at locating and exploiting these pockets.

---

> > ### Comment · Reviewer_cTGP · 2025-06-11
> >
> > Thank you for the additional analysis and experiments.
> >
> > Regarding the Initial Sampling Strategy, the authors’ response has addressed some of my concerns.
> >
> > However, the experiments on Error Diversity and Comparison with ACD do not fully address my concern. Since the 13 categories are predefined by the knowledge base and the initial batch, I am actually more interested in the diversity of errors within each category. I suggest that showing some specific cases of errors discovered by ACD and SEA might allow for a more intuitive and direct comparison.
> >
> > The experimental results of the scale of the knowledge base are interesting. It seems that under a fixed budget, SEA discovers a similar number of errors regardless of the knowledge base size. this might be due to the model’s inherent deficiencies or to the query budget, which limits the number of errors. However, this also suggests that simply increasing the size of the knowledge base does not lead to clear gains for SEA. Additionally, the statement “The number of errors discovered by SEA is not limited by the size of the knowledge base” to be somewhat imprecise. For instance, if the knowledge base only contains one item, SEA clearly would not be able to discover many errors.
> >
> > I encourage the authors to include further experiments and analysis in a revision.
> >
> > Overall, I appreciate the authors’ response, which addressed some of my concerns. I feel my overall score falls between 5 and 6—perhaps 5.5. Given the system only allows for integer scores, I will slightly bump my score up to 6.

---

### Decision · Program_Chairs · 2025-07-08

**Decision:**

Accept

**Comment:**

This paper proposes Stochastic Error Ascent (SEA), a framework for efficiently discovering knowledge deficiencies in large language models (LLMs) under a limited query budget. SEA combines semantic similarity retrieval and graph-based error propagation methods to iteratively uncover model deficiencies without accessing internal weights. It significantly outperforms baselines in error detection and cost efficiency, though some concerns remain about bias, error diversity, and implementation details.